# Vertical Profiles of Pollution Particle Concentrations in the Boundary Layer above Paris (France) from the Optical Aerosol Counter LOAC Onboard a Touristic Balloon

**DOI:** 10.3390/s20041111

**Published:** 2020-02-18

**Authors:** Jean-Baptiste Renard, Vincent Michoud, Jérôme Giacomoni

**Affiliations:** 1LPC2E, CNRS / Université d’Orléans, 45071 Orléans CEDEX 2, France; 2LISA, CNRS / Université Paris-Est-Créteil, Université de Paris, Institut Pierre Simon Laplace (IPSL), 94010 Créteil CEDEX, France; Vincent.Michoud@lisa.u-pec.fr; 3Aerophile SAS, 75015 Paris, France; giacomoni@aerophile.com

**Keywords:** particulate matter, urban pollution, number concentrations, tethered balloon

## Abstract

Atmospheric pollution by particulate matter represents a significant health risk and needs continuous monitoring by air quality networks that provide mass concentrations for PM10 and PM2.5 (particles with diameter smaller than 10 μm and 2.5 μm, respectively). We present here a new approach to monitor the urban particles content, using six years of aerosols number concentration measurements for particles in the 0.2−50 μm size range. These measurements are performed by the Light Optical Aerosols Counter (LOAC) instrument onboard the tethered touristic balloon “Ballon de Paris Generali”, in Paris, France. Such measurements have allowed us first to detect at ground a seasonal variability in the particulate matter content, due to the origin of the particles (anthropogenic pollution, pollens), and secondly, to retrieve the mean evolution of particles concentrations with height above ground up to 150 m. Measurements were also conducted up to 300 m above ground during major pollution events. The vertical evolution of concentrations varies from one event to another, depending on the origin of the pollution and on the meteorological conditions. These measurements have shown the interest of performing particle number concentrations measurements for the air pollution monitoring in complement with regulatory mass concentrations measurement, to better evaluate the intensity of the pollution event and to better consider the effect of smallest particles, which are more dangerous for human health.

## 1. Introduction

Atmospheric pollution by particulate matter (PM) is a growing concern, particularly in urban environments that concentrate a large portion of the population and the particle’s emission sources. These particles can be primary, directly coming from natural sources (dusts, salts, pollen) and from anthropogenic sources (transport, heating, industries, agriculture) or secondary, coming from chemical reactions involving sun light or atmospheric oxidants.

Such particles represent a significant health risk [1,2,3]. The PM 10 fraction (particles with aerodynamic diameter smaller than 10 μm) can penetrate beyond the nasopharyngeal tract in the bronchi and up to the pulmonary alveoli. The smaller particles, below 1 μm, can diffuse in the body and be found in various human organs [4,5,6]. The smaller the particles are, the deeper they can penetrate and diffuse in the body. Thus, detecting and counting the submicronic pollution particles is a major area of interest for public health. Exceeding the WHO (World Health Organization) guideline values for PM2.5 (particles with aerodynamic diameter smaller than 2.5 μm) of 10 μg·m^−3^ is the cause for about 20,000 premature deaths or more each year in major European cities [7]. In particular, it has been estimated that if these standards would have been reached it would have resulted in a gain of six months of life expectancy for more than 11 million inhabitants of the Paris region [8].

The Paris region is characterized by relatively few industries. At a regional scale, the urban background conditions are driven by long-distance transport of pollutants, accounting for 70% in average of PM2.5 mass-concentration [9,10,11,12,13]. Local sources of primary particles are dominated in mass by traffic emission and by residential heating in winter, as in many other urban environments [14,15]. The spatial distribution of particles concentrations at the surface can strongly differ at the urban scale, also depending on air-mass dispersion constrained by the urban topography.

At ground, the PM content is continuously monitored by air quality networks, which provide the mass concentrations of PM10 using microbalances or similar instruments. Some stations can also provide an estimate of PM2.5 mass concentrations. In Paris and the “Ile de France” region (which includes the city of Paris) these regulatory measurements are conducted by Airparif [16]. However, the techniques deployed across the air quality monitoring networks are by law oriented toward the measurement of the aerosol mass concentration, while it is known that the smallest particles, which penetrate deeper into the human organism, mostly dominate the number concentration and contribute only weakly to the mass. Then, complementary measurements providing particle number concentrations in addition to mass concentrations can be proposed to monitor the urban particle content and better understand its distribution and dynamics. Vertical profiles measurements of the aerosols content above major cities, performed during specific campaigns, can be used to better understand the vertical transport and the dispersion of the particles [17,18,19].

The aim of this paper is to present six years of aerosols number concentration measurements performed by the Light Optical Aerosols Counter (LOAC) instrument onboard a tethered touristic balloon in Paris. Such conditions of measurements allow us to better determine the mean seasonal variations of particles content for different size classes, and also to evaluate the mean vertical evolution of particles and to focus on some specific pollution events.

## 2. The Light Optical Aerosols Counter (LOAC)

The LOAC is a prototype instrument developed for ground and balloon-based measurements [20]. Particles are drawn up to the optical chamber through an isostatic tube by a small pump and cross a laser beam working at 650 nm. The scattered light is recorded by two photodiodes at scattering angles of ~15° and ~65°, and photons travel directly to the photodiodes though pipes without a lens (Figure 1, updated from [21]). A total of 19 size classes are defined in the particle diameter range between 0.2 and ~50 μm. The size classes are chosen as a good compromise between the instrument sensitivity and the expected size distribution of ambient air particles. For a 10 min integration time, the uncertainty of total concentrations is about ±20% for concentrations higher than 10 particle·cm^−3^ up to ±60% for concentrations lower than 10^−1^ particle·cm^−3^. The uncertainties in the size determination is of ±0.025 μm for particles smaller than 0.6 μm, 5% for particles in the 0.7−2 μm range, and of 10% for particles greater than 2 μm.

The measurements at 15° are almost insensitive to the refractive index of the irregular shaped aerosol particles and can be used to determine the concentrations [22]. On the other hand, measurements at side-scattering around 65° are very sensitive to the refractive index of the irregular particles [23]. An indication of the typology of such particles can be obtained using a “speciation index”, retrieved by combining the 15° and 65° channels measurements. This index is sensitive to the imaginary part of the refractive index of the particles, and thus to their optical absorbing properties. Laboratory references for the speciation indexes have been determined with LOAC for 4 natures of particles: carbonaceous, mineral dust, salts, and liquid droplet [20]. The speciation indexes obtained from LOAC observations in the ambient air are compared to these laboratory data to derive the estimated dominant typology of particles in different size classes. The identification of the nature of the particles works well in case of a homogeneous medium but is more questionable in case of a heterogeneous medium that cause the speciation index to be more scattered. 

Finally, the concentrations can be converted to mass concentrations (in μg·m^−3^) assuming spherical particles and mass density between 1.2 and 2.2 g·cm^−3^ depending on the detected typology of the particles. For carbonaceous particles smaller than 1 μm, the mass density is assumed to be 2.0 g·cm^−3^, while the density is assumed to be of 1.2 g·cm^−3^ for particles greater than 1 m. For mineral particles the density is assumed to be of 2.2 g·cm^−3^; in case of no identification, the density is assumed to be of 2.0 g·cm^−3^ (these values are updated from [20]). The error bars are calculated considering the uncertainties in size determination. The LOAC average mass-concentrations accuracy is of about ±5 μg·m^−3^ when compared to microbalance measurements in laboratory [20].

The performances of LOAC have been established during numerous sessions of inter-comparison with other instruments dedicated to the counting, size distribution, extinctions, and mass concentrations of solid aerosols in the atmosphere [20]. These sessions have shown that LOAC can be used for studies on urban aerosols. 

## 3. Measurements Conditions at the Touristic “Ballon de Paris Generali” in Paris (France)

The LOAC has been mounted on the gondola of the touristic tethered balloon “Ballon de Paris Generali” (Figure 2) in the park “André Citroën” in the south-west of Paris, France (48.8414°N, 2.2740°E), since mid-2013 and works continuously [24]. Depending on the meteorological conditions, 150 to 200 days per year are favorable for flying. The height is measured by a GPS, with a vertical accuracy of about ±15 m. The balloon nominal maximum height above ground is 150 m and up to 50 flights can be performed per day. Some flights can also be conducted up to a height of 300 m when the wind speed is very low (< few m·s^−1^). The urban pollution events occur mainly during anticyclonic conditions when the wind speed is low; thus, the balloon can often fly during such events.

Due to the balloon location in one of the largest parks in Paris with no main highways in the direct vicinity, the ground-based measurements can be referred as background urban conditions (Figure 3). The highest buildings close to the park have a height below 50 m. Therefore, we can expect to measure mean persistent urban pollution when the balloon height is above 50 m. Nevertheless, these measurements are conducted just at one location in Paris, thus the results presented below could be not representative of the general picture. 

LOAC provides measurements every 10 s. During the balloon flight, data must be integrated over at least 30 s to reduce the measurement noise. Since the balloon ascent speed is of 1 m·s^−1^, the vertical resolution for an individual LOAC concentrations profile is of about 30 m. Individual profiles can be also daily averaged to reduce the natural dispersion of the concentration’s measurements. For measurements at ground, the data can be averaged over 10 min to reduce the dispersion of the measurements.

## 4. Measurements at Ground

Figure 4 presents the 2013−2019 evolution of the particle number concentrations with time for the 19 size classes of LOAC when the balloon is at ground and outside major LOAC and/or balloon maintenance period. Data during fog and heavy rain events have been removed because of the presence of droplets that can skew the retrieval of the solid particles. For clarity reason, the data have been smoothed by a sliding smoothing procedure with a width of 2 days. The highest concentrations corresponding to strong pollutions events were recorded at the beginning of winter 2013, and at the beginning of 2014 and 2015. It can be noticed that the mean level of concentrations for particles smaller than 10 μm measured by LOAC is higher in the 2013 period than after (Figure 4), due to strong building activities close to the Parc André Citroën (unfortunately no Airparif station is close to this location to validate these observations).

The PM10 mass concentrations obtained by the Airparif air quality network for a station in the suburb of Paris (Vitry, background urban conditions, 48.7778°N, 2.3779°E) and for a station in a rural area in the south of Paris region (Rural South, 48.3667°N, 2.2333°E) also show the concentrations enhancements during pollution events (Figure 5). The mass concentrations derived from the LOAC measurements are also plotted in Figure 5 and are globally in good agreement with the Airparif measurements (all the data have been also smoothed with the same procedure as for LOAC number concentrations). Excluding the 2013 period with the building activities close to the park, the mean LOAC mass concentrations measurements are 6.5 μg·m^−3^ and 0.5 μg·m^−3^ lower than the Vitry and Rural South stations, with a standard deviation of 10 μg·m^−3^ and 9.5 μg·m^−3^ respectively.

When considering the six years of measurements, a seasonal cycle seems to be present for the particles smaller than about 1 μm, with maximum concentrations in winter and minimum concentrations in summer (Figure 6). This could be related to the heating and traffic during winter but also to the seasonal variation of the boundary layer height [25] and to the meteorological conditions. The concentrations for particles larger than about 15 μm exhibit also a seasonal cycle, anticorrelated with the seasonal cycle of the smaller particles; the highest concentrations are obtained during summer (Figure 6). A possible explanation is the detection of pollens, since most of the pollens are indeed in the 15−50 μm size range and their seasonality is similar to the one detected by LOAC.

## 5. Vertical Profiles

### 5.1. Daily Profiles

During the 2013−2019 period, 976 days with flights performed during daytime are available. Among them 148 days with flights up to the height of 300 m were performed. Since most of the flights were conducted up to the height of 150 m, we will consider these flights to estimate the mean evolution of the concentrations with height. On the other hand, the few flights per month up to the height of 300 m can be used to study some specific events.

To study the mean vertical trend, the concentrations are integrated over 7 height ranges: 0−20 m, 20−40 m, 40−60 m, 60−80 m, 80−100 m, 100−120 m, and 120−150 m. The vertical profiles are daily averaged to reduce the variability of the individual measurements. This variability originates from the measurement accuracy (Poisson distribution) but also from variations in the wind conditions. Globally, the daily standard deviation of the concentration profiles is in the 20%−60% range. Thus, the daily profiles presented in the following subsections contains error bars that represent the standard deviation for each size class and height range.

### 5.2. Background Conditions

The daily profiles exhibit a temporal variability of about two orders of magnitude depending on the weather and on the pollution conditions. As examples, Figure 7 presents two number concentrations profiles obtained in summertime, 8 July 2016 and 20 September 2017, during low pollution conditions (the LOAC PM10 mass concentrations were of about 10 μg·m^−3^ at ground for both cases and in the 5−15 μg·m^−3^ range in flight). The two profiles exhibit different vertical evolutions and size distributions; in particular, the profile of the 8 July 2016 presents an excess of large particles in the 10−30 μm size range, probably due to a pollen episode.

At each height range, the histogram of the number concentrations obtained during the six years of measurements do not follow a gaussian distribution. On the other hand, a gaussian distribution can be obtained when considering the logarithm of the concentrations. Four broad-range classes are defined: 0.2−1.0 μm, 1−3 µm, 3−10 μm, and 15−50 μm. Then, for each height range and for the 4 broad-range classes, the histogram of the logarithm of the concentrations is fitted by a gaussian function, to estimate the mode of number concentrations (corresponding to the most frequent concentrations over the full time period of measurements, examples are given in Figure 8).

Figure 9 presents the evolution with height of the more frequent concentrations for the 4 broad-range classes. It must be noticed that these vertical profiles might not be individually consistent and are not the modal vertical profiles but are a sequence of individual modal concentration for each height range. For the submicronic particles, no evolution is detected, while the concentrations slightly decrease of about 20% in the first tens of meters for the largest particles. This could result of the particles sensitivity to upward diffusion, depending on their sizes, shapes, and densities, but this analysis requires further investigation.

### 5.3. Main Pollution Events

During most of the pollution events, the winds were low, thus flights up to a height of 300 m were available. We present below the main pollution events observed by LOAC in the 2013−2019 period.

#### 5.3.1. Winter Event

Pollution is often encountered in Paris during anticyclonic conditions in winter, typically in December, with particulates originating from traffic and heating [26]. During such events, the concentration of the particles can be from 10 to 100 times higher than during the summer background conditions. The stronger winter event recorded in the 2013−2019 period by LOAC occurs in December 2013. Figure 10 presents the profiles for the strong pollution event in 11 December 2013. The profile exhibits a strong concentration increase at a height of 200 m for the smallest particles <0.4 μm while the concentrations are about ten times lower above, due to a temperature inversion layer resulting from the anticyclonic conditions. The LOAC PM10 mass concentrations at ground are of about 50 μg·m^−3^ at ground, of about 60 μg·m^−3^ at a height of 130 m due to an increase of concentrations of particles in the 5−10 µm size range, only of 45 μg·m^−3^ at a height of 200 m although the increase of submicronic particles concentrations, and of about 35 μg·m^−3^ above. Thus, the height variability in mass concentrations do not fully reflect the more complex height variability of the number concentrations for the various size classes.

#### 5.3.2. Spring Event

Another type of pollution is often encountered in beginning of spring, typically in March, mainly originating from the combination of local pollution and the transport of pollutants from agricultural activities around Paris [27,28]. The vertical profiles in Figure 11 for the 14 March 2014 and 17 March 2015 differ from those measured in winter (Figure 10). The concentrations of the particles are more constant with height, because boundary layer is thicker due to more vertically mixed air masses, while concentrations of the biggest particles slightly decrease with height. The LOAC PM10 mass concentrations decrease from 110 μg·m^−3^ at ground to about 65 μg·m^−3^ at 300 m for the 17 March 2015 and decrease from about 40 μg·m^−3^ at ground to about 20 μg·m^−3^ at 300 m (Figure 12).

#### 5.3.3. Summer Event

Pollution events can be also present during summer, depending on the weather conditions. Although no major pollution event was detected at ground during the 2014−2019 period, an increase in aerosol concentrations was detected from 24 to 30 June above a height of about 100 m. For the 25 June 2019 profile, the number concentrations were up to 10 times higher for particles smaller than 5 μm in the 100−200 m height range than at ground, the mass concentration were five times higher (Figure 13). This event was not detected from ground-based station, showing the interest of balloon-borne measurements.

No explainable extra-source (smoke, industrial activities) was identify during this period at the vicinity of the measurement location. The LOAC typology indication differs strongly from those of the other pollution events and do not correspond to the usual LOAC typology for conventional urban pollution particles (carbonaceous, mineral, sulfate, ammonium nitrate). At present no explanation can be proposed for the origin of this event and for the nature of the particles, and further investigations are needed.

## 6. Discussion

The concentrations of the particles and their vertical evolution varies from one pollution episode to another. To better evaluate the variability, we can consider the previous 4 broad-range classes, now applied to the 6 vertical profiles presented in Figure 7, Figure 10, Figure 11 and Figure 13. A factor of more than 50 occurs between low pollution and high pollution events for the 3 first broad-range classes (particles with diameter smaller than 10 μm), as shown in Figure 14. As expected, the highest concentrations of the submicronic particles occur for the December 2013 event and for the March 2014 and 2015 events. For the particles in the 1−3 μm size range, the highest concentrations were detected during the March 2014 event, during the March 2015 event, and during the June 2019 event in flight. In particular, the March 2015 event was dominated by secondary aerosols (more particularly of ammonium nitrate, as reported in [28]). The concentration of the 1−3 μm particles during the December 2013 event is significantly lower than for the March 2014 event. The tendency for the 3−10 μm particles is relatively similar to the one for the 1−3 μm particles, except for the March 2015 concentrations that are closer to the background conditions. Finally, no obvious correlation between pollution and concentrations of the largest particles (15−50 μm) can be pointed out, since, as previously mentioned, these particles are likely mainly originating from pollen events.

When considering the mass concentrations values from LOAC, the highest pollution events were on March 2014 and March 2015 (with maximum values up to 110 μg·m^−3^ and up to 90 μg·m^−3^, respectively). When considering the number of all particles larger than 0.2 μm, the highest concentrations were detected on December 2013, although the mass concentrations were only of up to 80 μg·m^−3^. These differences could be due to the origin of the pollution, typically primary (carbonaceous) particles versus secondary aerosols.

Mass concentration variability from low pollution to high pollution levels in Paris is about one order of magnitude, while the variability is of two orders of magnitude for number concentration. Most of the studies on the pollution trend in cities are based on mass concentrations [29]. It is often established that the mass-concentration of PM10 is decreasing in European cities, mainly due to changes in diesel engines and particles filters [30]. Nevertheless, possible change in the size distribution, as possible increase in number concentration of submicronic particles, cannot be estimated with such measurements. Also, the vertical evolution of particles number concentrations can change depending on the origin of the pollution events. Number concentration measurements as those presented above, mainly for submicronic that can be the most dangerous for health [4,5,6], may be more efficient than mass concentrations to evaluate the health impact of pollutions events and to establish the temporal trend of number concentration of (anthropogenic) pollution particles. Obviously, such trends cannot be retrieved from measurements only at one location; thus, a network of (optical) counter instruments must be implanted in parallel with regulatory mass concentration instruments.

## 7. Conclusions

The six years of number concentration measurements performed in Paris, France, by the LOAC instrument for particles in the 0.2−50 μm size range allowed us to detect a seasonal variability in the PM content, expectedly due to the origin of the particles (anthropogenic pollution, pollens). Since the instrument is mounted onboard the touristic tethered balloon “Ballon de Paris Generali”, the mean evolution of particles concentrations with height was obtained for background conditions up to a height of 150 m. The concentrations of submicronic particles remain constant with height, while the concentrations slightly decrease in the first tens of meters for the largest particles. Measurements were also conducted during major pollution events, with particles concentrations several tens of times higher than during background conditions. The vertical evolution of concentrations varies from one event to another one, depending on the origin of the pollution and on the meteorological conditions. In particular, an accumulation layer was sometimes detected at a height of about 200 m in winter due to temperature inversion layer.

These measurements have shown the interest of performing particle number concentrations measurements for the air pollution monitoring in complement with regulatory mass concentrations measurements, to better evaluate the intensity of the pollution event and to better consider the effect of smallest particles.

We have presented here the first results obtained with LOAC at the Paris touristic balloon. Future studies could be conducted using this LOAC database, as modeling works on the vertical transport of particles, studies on the correlation of the variability of the particle’s concentrations and their size distribution with the meteorological conditions, and studies on the evolution of the particle concentrations after rain. Finally, a LOAC is also mounted in another touristic balloon, the “Ballon Terra Botanica” in a smaller city, Angers (West of France); its measurements will be compared soon to those of Paris to better establish the similarities and the differences for the vertical evolution of the pollution particles depending on the measurement locations.

New measurements could be proposed by mounting the LOAC instrument onboard the various touristic tethered balloons available in the world, to better evaluate the vertical transport of the urban pollution particles.

## Figures and Tables

**Figure 1 sensors-20-01111-f001:**
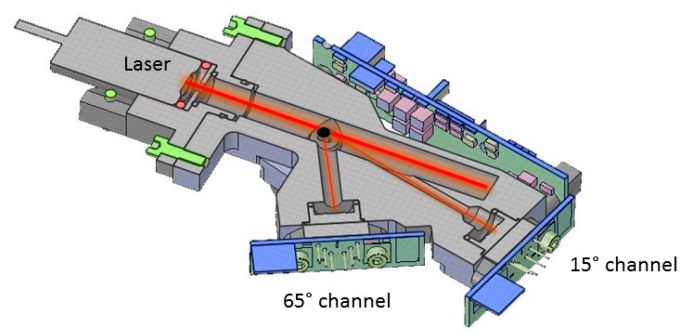
Light Optical Aerosols Counter (LOAC) principle of measurements at two scattering angles (updated from Renard et al., 2018).

**Figure 2 sensors-20-01111-f002:**
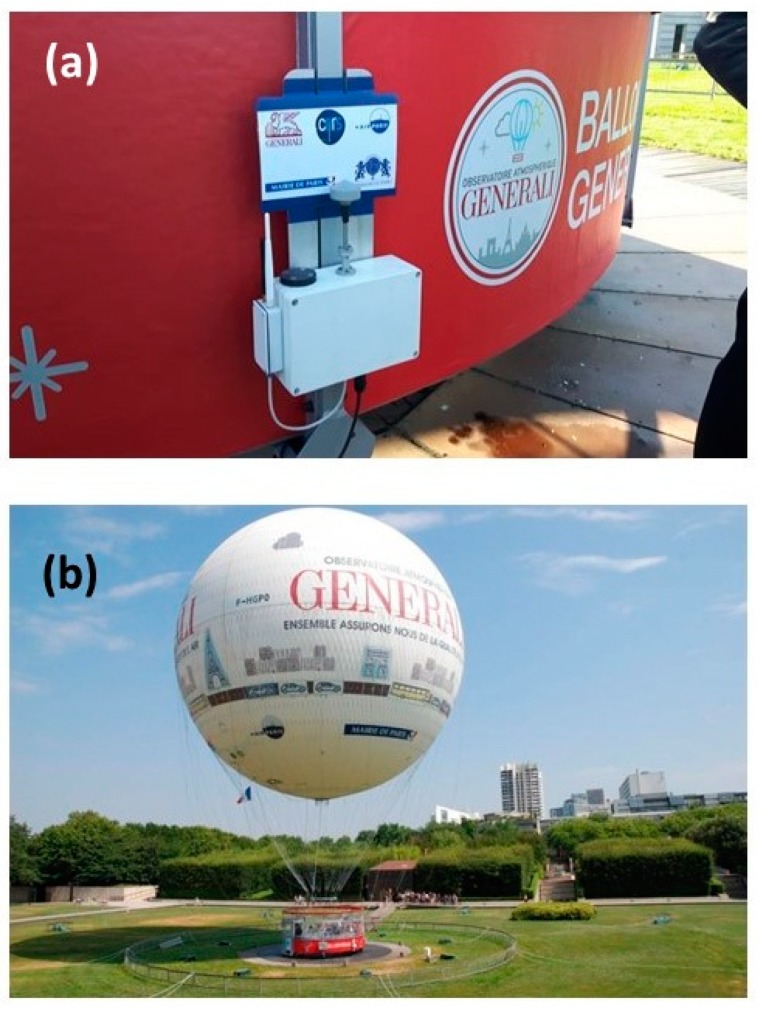
(**a**) LOAC mounted on the gondola of the “Ballon de Paris Generali”; (**b**) The balloon in the park André Citroën (Paris, France).

**Figure 3 sensors-20-01111-f003:**
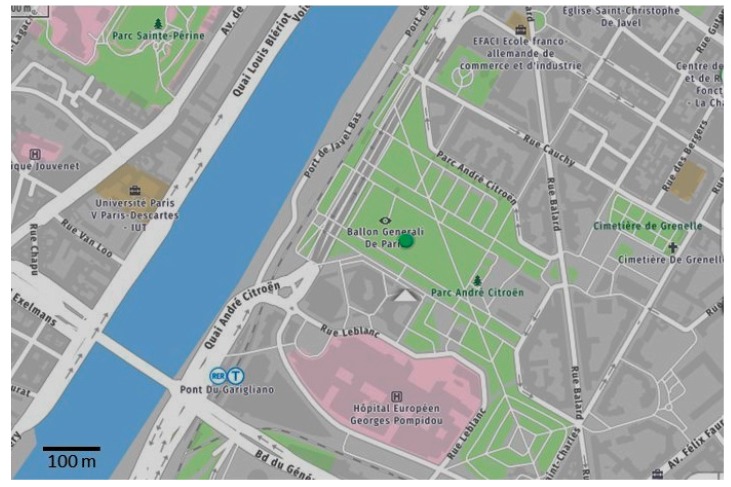
Map of the vicinity of the “Parc André Citroën” (the green dot represents the balloon location, map from mappy.com).

**Figure 4 sensors-20-01111-f004:**
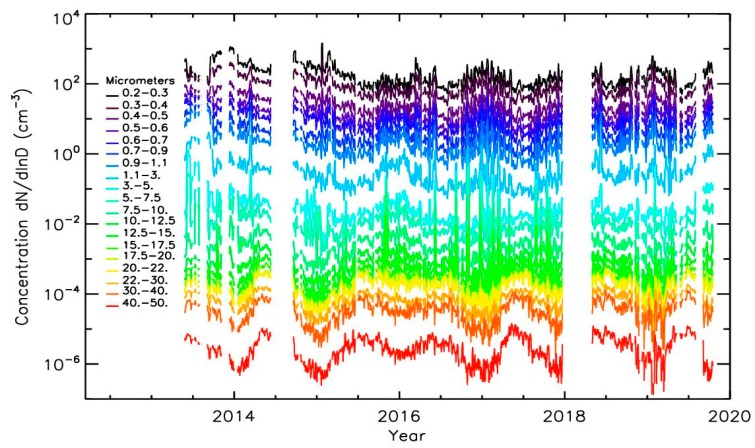
Temporal evolution of number concentrations for the 19 LOAC size classes when the balloon is at ground (the data have been smoothed by a sliding smoothing procedure with a width of 2 days).

**Figure 5 sensors-20-01111-f005:**
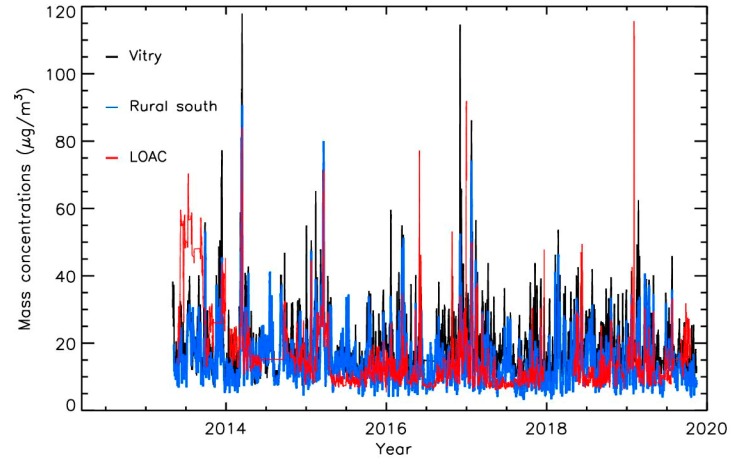
Temporal evolution of the PM10 mass concentrations from LOAC at the “Ballon de Paris Generali” and from the Airparif air quality network for stations in the suburb of Paris (Vitry) and in the rural area in the south of Paris region (the data have been smoothed by a sliding smoothing procedure with a width of 2 days).

**Figure 6 sensors-20-01111-f006:**
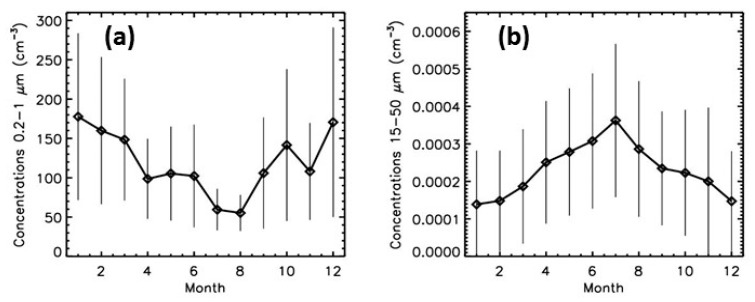
Annual evolution of the concentrations for the smallest particles (**a**) and the biggest particles (**b**) detected by LOAC for the six-year period; the vertical bars correspond to the mean absolute deviation of the concentrations.

**Figure 7 sensors-20-01111-f007:**
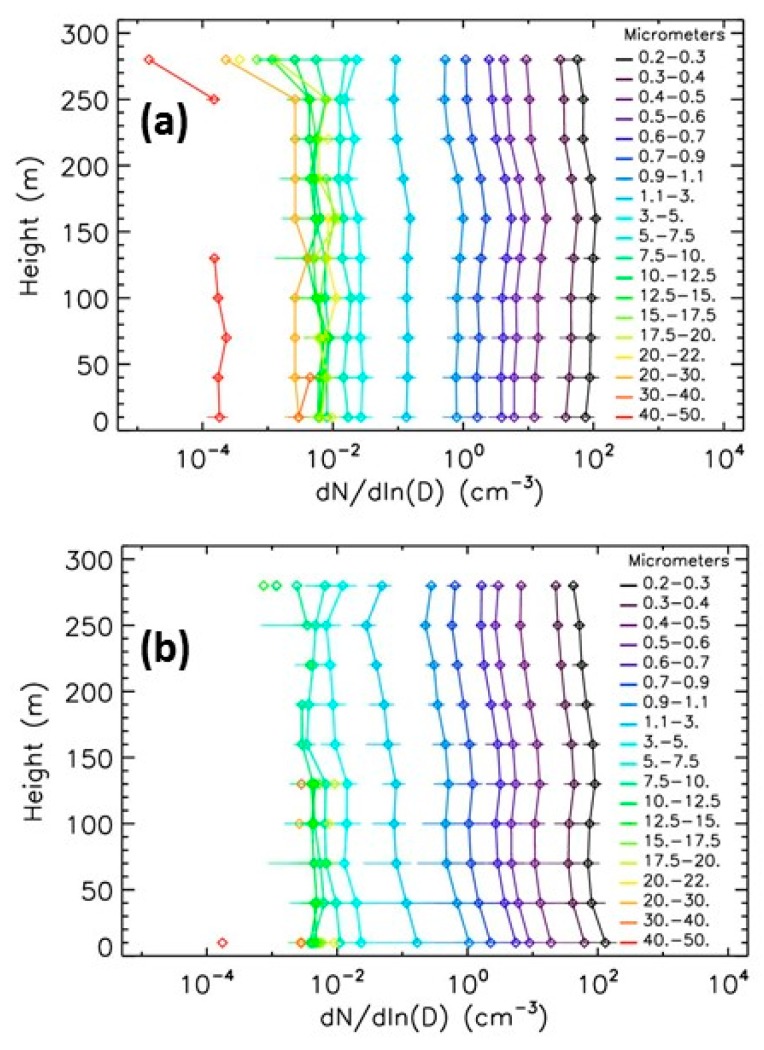
Evolution with height of number concentrations for the 19 size classes of LOAC when the urban pollution is low. (**a**) 8 July 2016; (**b**) 20 September 2017.

**Figure 8 sensors-20-01111-f008:**
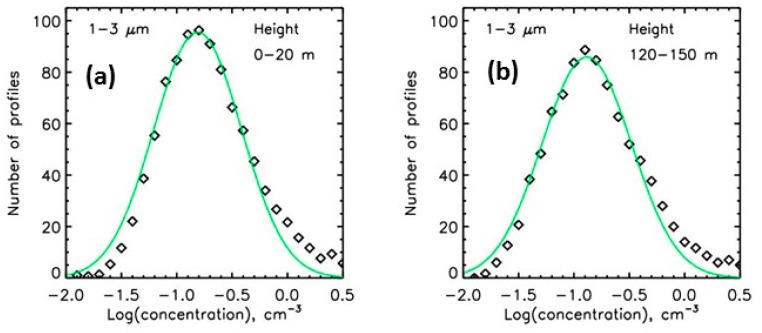
Examples of the histogram of the logarithm of the concentrations at two height ranges for the six years of measurements, for the 1−3 μm size range. (**a**) measurements at ground; (**b**) measurements at the height of 120−150 m.

**Figure 9 sensors-20-01111-f009:**
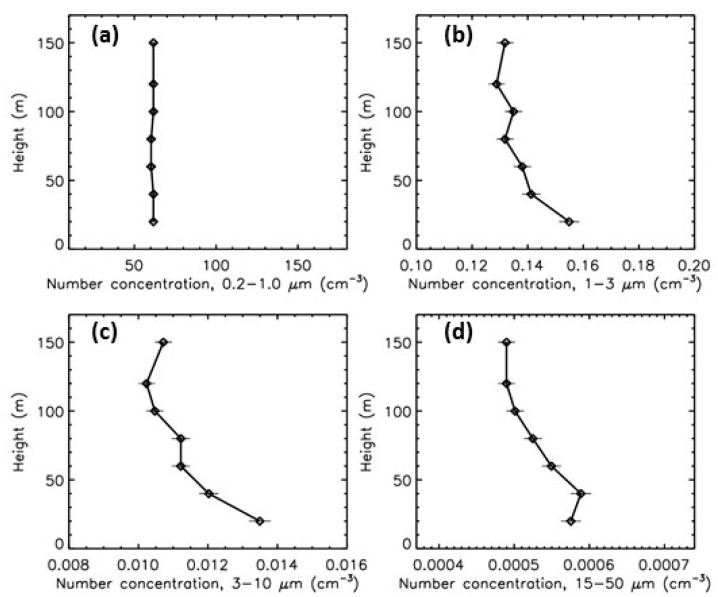
Evolution with height of the mode of number concentrations for the six years of measurements. (**a**) broad-range class 0.2−1.0 µm; (**b**) broad-range class 1−3 µm; (**c**) broad-range class 3−10 µm; (**d**) broad-range class 15−50 µm.

**Figure 10 sensors-20-01111-f010:**
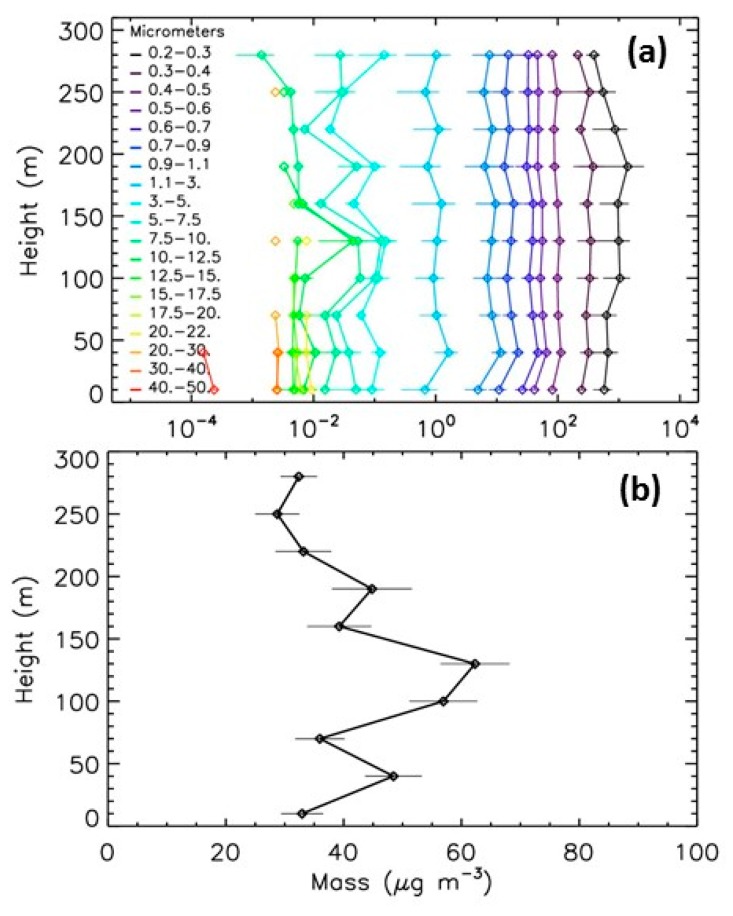
Evolution with height of LOAC concentrations for the 11 December 2013 pollution event. (**a**) number concentrations; (**b**) PM10 mass concentration.

**Figure 11 sensors-20-01111-f011:**
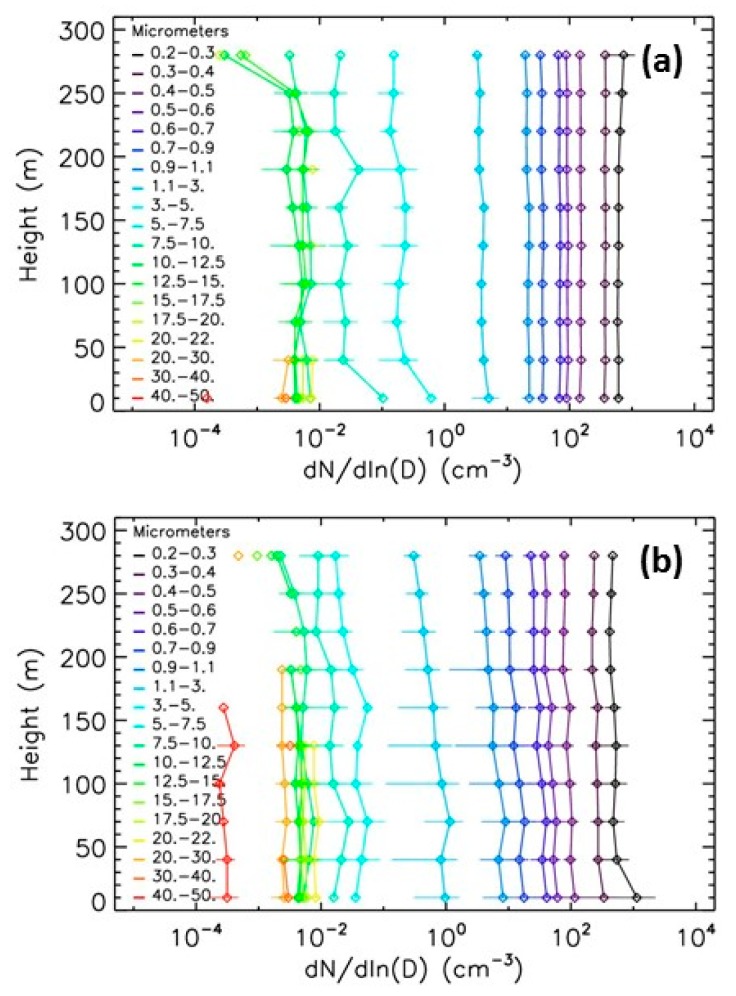
Evolution with height of LOAC number concentrations during the spring pollution events. (**a**) 14 March 2014; (**b**) 17 March 2015.

**Figure 12 sensors-20-01111-f012:**
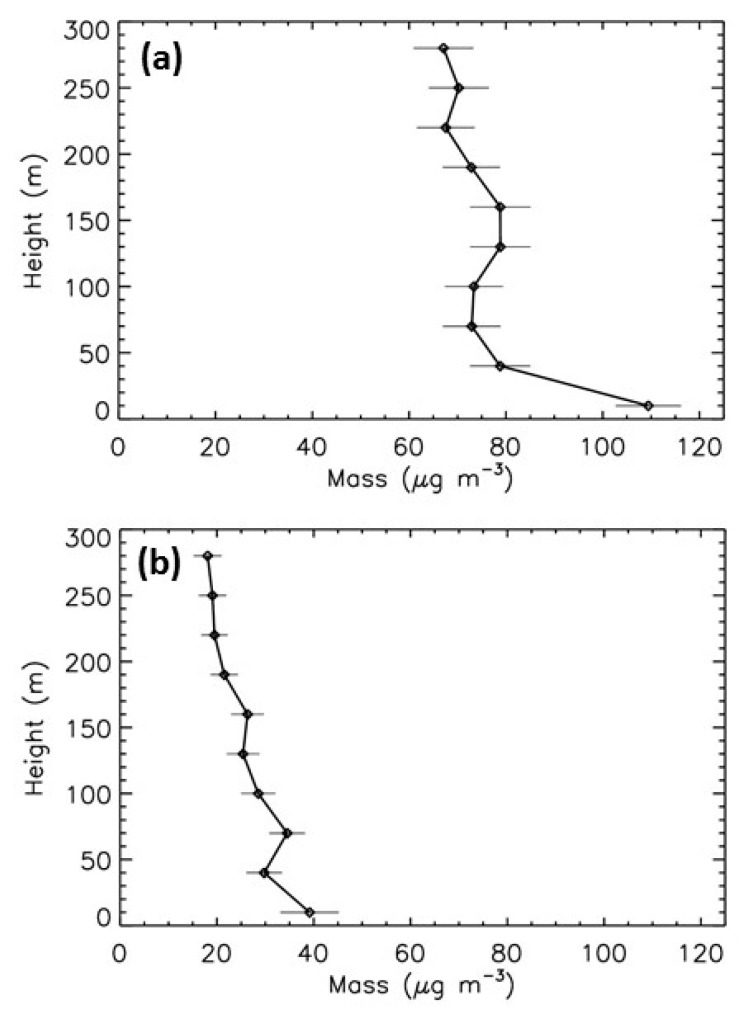
Evolution with height of LOAC PM10 mass concentrations during the spring pollution events. (**a**) 14 March 2014; (**b**) 17 March 2015.

**Figure 13 sensors-20-01111-f013:**
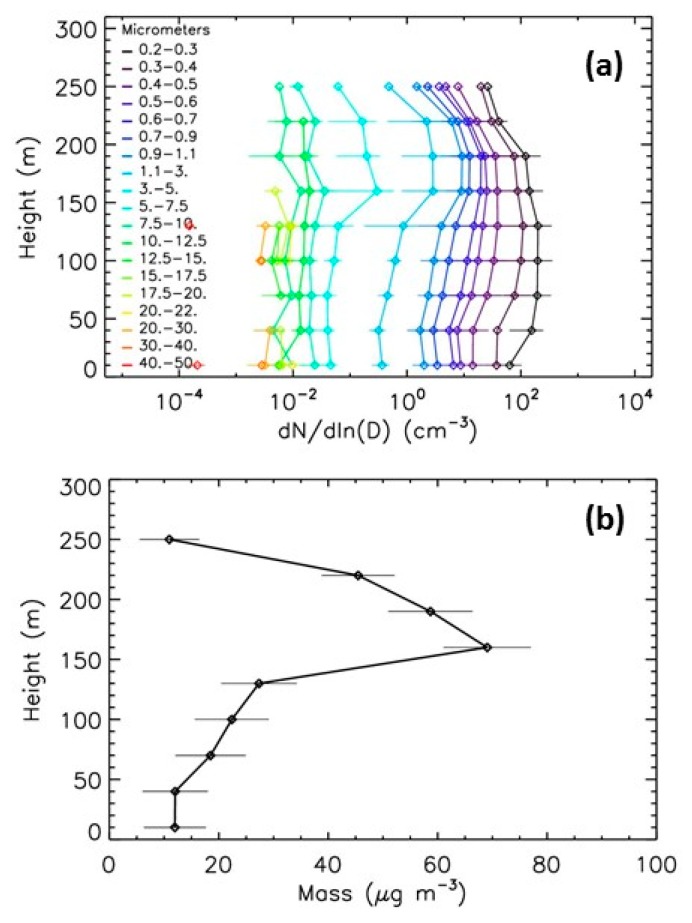
Evolution with height of LOAC concentrations for the 25 June 2019 pollution event. (**a**) number concentrations; (**b**) PM10 mass concentration.

**Figure 14 sensors-20-01111-f014:**
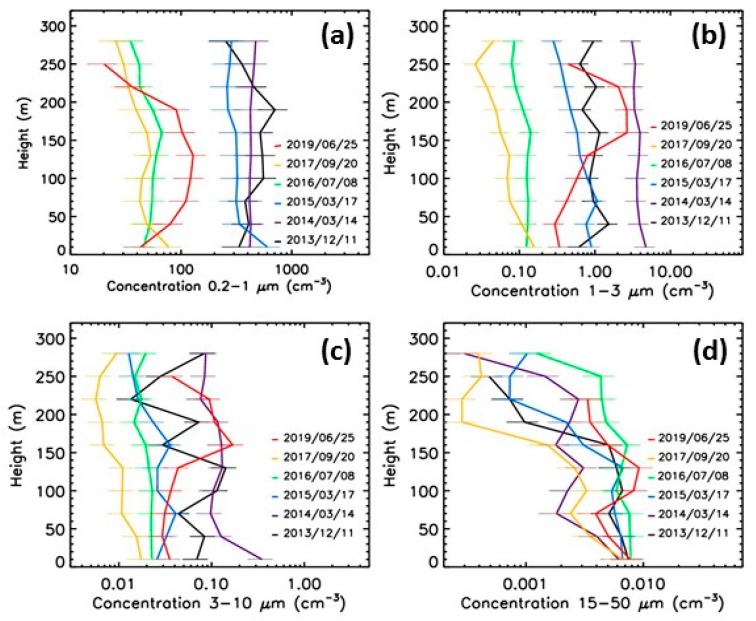
Evolution with height of aerosols concentrations. (**a**) broad-range class 0.2−1.0 µm; (**b**) broad-range class 1−3 µm; (**c**) broad-range class 3−10 µm; (**d**) broad-range class 15−50 µm.

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
