# Peer review of "Vertical Profiles of Pollution Particle Concentrations in the Boundary Layer above Paris (France) from the Optical Aerosol Counter LOAC Onboard a Touristic Balloon"

_sensors, 2020, doi:10.3390/s20041111_

Round 1
Reviewer 1 Report
The paper is interesting and should be published however there are some matters in the text that need attention as follows:
There is no discussion of how the vertical profiles change during the day as meteorological conditions vary and therefore whether it is appropriate to use daily profiles. What is the variability form one profile to the next? This needs discussion. Lines 244-245: ‘probably because of their sedimentation’; surely this just reflects local sources since the gradients are consistent with upward diffusion. The particle sizes considered have small sedimentation velocities. This also needs appropriate discussion. Summer event. The elevated maximum suggests an upstream elevated source (a fire?). Was the wind direction consistent with any notable upstream source? Line 322. Accumulation seems highly implausible. Surely an upstream source?
Reviewer 2 Report
The manuscript provides an overview of the vertical and temporal variability in particle number size distribution (PNSD) over Paris from the second half of 2013 until nowadays. The authors describe the methods and the instruments used, one representative pattern for both low and high pollution conditions, a modal vertical profile for PNSD and PM10, few sample patterns for winter, spring and summer. The authors finally draw some conclusions based on the main findings in the analysis above.
The collected dataset is certainly unique and with a great potential, and an introductory work is surely of interest by the scientific community. The language is somewhat poor and the text is not always clear.
One main preliminary comment for the language: “altitude” is the height above mean sea level. So all “altitude”s in the manuscript must be changed to “height” or “height above ground”.
The authors attributed all occurrences of increase in number concentration for larger particles to pollen. Do they have some way to prove it? Why can’t it be soil, dust or particles other than pollen? The peak for larger particles is in July: is it this consistent with the seasonality of expected pollen in the park hosting the tethered balloon?
How do the authors measure the height above the ground? Is there any meteorological variable associated to these PNSD data?
It is hardly possible to compare these profiles without applying a normalisation for each profile according to some variable describing atmospheric mixing, e.g. potential temperature. In order to get any solid understanding about patterns this normalisation is strongly encouraged. A good examples can be found in Ferrero et al., 2016 (doi:10.5194/acp-16-12601-2016) and more indications in the references therein.
Some extra minor comments regarding the manuscript follows:
p.1 line 40: “Smaller are the particles, deeper they can penetrate and diffuse in the body.”. According to main literature this is wrong (besides that it should be “the smaller …, the deeper”). Assuming the smallest particle to be 1nm in “diameter” (the quotes are on purpose), main respiratory deposition models show that the particles smaller than 10-20 nm (i.e. the smallest) diffuse to trachea, bronchi and head airways, and they don’t reach to the alveoli.
p.2 line 52: “local scale” in atmospheric science usually indicates domains of few kilometers. Did you mean “urban scale”?
p.2 lines 74, 86: forward scatter is collected at 15°, but in figure 1 it is declared to be 12°
p.7 line 200: the authors are log-transforming particle counts in order to have a Gaussian distribution of concentration. Then they fit the data to find the modal logarithmic concentration. Then, the reader needs to assume (since this is not explained in text), that the authors applied an inverse transformation to derive the modal concentration in “standard” units. And this operation is done for 4 size ranges and for 7 heights ranges. This procedure is quite involved, and looking at the data presented I wonder why the authors did not simply use the most frequent concentration for each case, without log-transforming the data. Besides this, the vertical profiles presented in figure 9, although extremely realistic, potentially they might have never been observed, since the authors analysed each height range separately: although this procedure is totally fine, if they authors intend to keep figure 9, they need to state clearly that those vertical profiles might not be individually consistent and those are not the modal vertical profiles, but a sequence of individual modal concentration for each height range.
p. 12 line 280: it would help the reader if the authors wrote to which cases the figures are referring to, i.e.: low and high typical pollution conditions, and spring and summer events.
p. 12 line 295: This sentence “the number of particles greater than 0.2 µm”, does it refers to particle between 0.2 and 1 µm or all particles larger than 0.2 µm?
p. 13 line 301: Fortunately for the scientific community this is wrong. It is true that the literature on trends for PM is large, simply because PM timeseries are more numerous and longer than particle number timeseries. Nonetheless there is some fundamental literature on trends in particle number since a decade, more or less. A classic paper is by Asmi et al., 2013 (https://doi.org/10.5194/acp-13-895-2013, 2013), which is also in the latest IPCC report. But there are several other studies on trends in particle number.
The language needs a large revision. Some minor indications are provided:
p.1 l.37: “particle with aerodynamic diameter”
p.1 l.39: “the smallest” or “smaller” particles? Maybe “smaller”
p.1 l.42: “particle whose aerodynamic diameter”
p.2 l.56: change “normative” to “regulatory”
p.2 l.72: change “though” to “through”
p.5 l.158: negative mass concentration are not possible. It’s stated already that the concentration are “lower”. No sign is needed
p.7 l.201, 202: maybe change “the more” to “the most”.
p.7 l.184: add “of magnitude” after “about two orders”. And do this everywhere it applies in the text
Figure 8: y-axis label is “Frequency” not “Number”
p.7 l.197: change “super-size class” to something meaningful like “broad-range class”
p.12 l.295: change “the higher” to “the highest”
p.13 l.302: change “PM10 are” to “PM10 is”
Please change all “counting concentrations” to “count concentration” or “number count concentration” or “particle count concentration” or something alike.
Reviewer 3 Report
“Vertical profiles of pollution particle concentrations in the boundary layer above Paris (France) from the optical aerosol counter LOAC onboard a touristic balloon” present here a new approach to monitor the urban particles content, counting measurements for particles in the 0.2-50m size range. These measurements are performed by the Light Optical Aerosols Counter (LOAC) to detect at ground a seasonal variability in the particulate matter content, due to the origin of the particles.
The amount of data is sufficient, and they are effective, traceable and persuasive. The method used is reliable with a theoretical basis and innovations. I just have some opinions on some problems in the article:
The author's novel point is that the Light Optical Aerosols Counter (LOAC) placed by balloon is used to continuously monitor PM2.5, PM10 and other pollutants. Combined with the actual pollution monitoring data, the author evaluates the intensity of pollution events and observes the formation and diffusion law of these pollutants. There is nothing wrong with this scheme for measuring particulate matter content and urban pollutant content, but I think one of the most important factors in designing a research topic is its feasibility and universality. Such data measurement results are only a small part of the very limited measurement results, the final need to be combined with the actual monitoring data, most of the cases can be directly used by the actual measurement structure. Secondly, such method has too many steps, too complicated process, and low cost performance in practice. Moreover, the data obtained in this way is not a very large sample size, so it is difficult to track for a long time, and it is less likely to provide support for the public (possibly a small part of people), which is not universal.

Round 2
Reviewer 2 Report
The authors made some minor changes, which only partially include the previous recommendation
In their reply the authors said “The Asmi et al. paper consider condensation particles counters (thus with no size distribution) and DMPS (having an upper limit of about 1 µm). None of the measurements were conducted in major cities. To our knowledge, it is the first time that timeseries of vertical profile in urban conditions at ground and in altitude are available. We may have been misunderstood. We do not speak of PM timeseries in general, but of PM mass concentrations measurements at ground during summer in Paris. We have changed “detectable” to “detected” to clarify our statement.”
And in the revised manuscript the authors wrote:
“Most of the studies on the pollution trend in cities are based on mass concentrations [26]. It is often established that the PM10 is decreasing in European cities, mainly due to changes in diesel engines and particles filters [27]. Nevertheless, no reliable results are available for tendencies of submicronic particles at ground. Also, the vertical evolution can change depending on the origin of the pollution events. Counting measurements as those presented above, mainly for particles smaller than 2.5µm that can be the more dangerous for health, may be more efficient than mass concentrations to evaluate the health impact of pollutions events and to establish the trend of (anthropogenic) pollution particles.”
My main comment follows.
When a reader reads “pollution trend in cities […] PM10 is decreasing [...]” assumes that the authors speak about a long term trend, i.e. a temporal trend. Then the authors claim that “ no reliable results are available for tendencies of submicronic particles at ground”, i.e. that there is no result for trend in particles lower than 1µm. Again this does not sound correct at all: Asmi et al. was an example including only remote sites, but it deals exactly with number concentration of particles with Dp < 1µm. Still there are several other articles on trends for number concentration of particles with Dp < 1µm: just to list a couple of the latest there is Sun et al., 2019, Atmospheric Environment or Sun et al. 2019, ACPD. These data proceeds from DMPS and include 20nm < Dp < 800nm. Papers on temporal trends on PM1 are indeed several and their results are totally reliable, moreover DMPSs are more stable than OPCs over time. And these are number concentration measurements, which are the type of data that the authors promotes in the same discussion. In case the authors in their reply mention that they are able to estimate temporal trends in vertical profile, they need to state it explicitly. And to support the possibility to estimate temporal trends, some information about the data coverage should be needed.
Please note that “Counting measurements” means that the authors have counted the measurements performed, but I guess you did not mean that.
“that can be the more dangerous for health”. This sentence includes at least 2 oversights: 1. may be you meant “the most dangerous” 2. assuming that point 1 is correct, then the authors should support this claim by including some reference explaining why particles with Dp < 2.5 µm should be the most dangerous for health, instead of those smaller than 3.5µm, or those smaller 100 nm or other diameters.
Finally the paragraph ends with “ to establish the trend of (anthropogenic) pollution particles” and the reader, based on the previous sentences, thinks about “temporal trends”, but apparently this is no more the case… here the authors meant some other type of trend, although they did not specify.
To wrap up, the authors first speaks about studies on temporal trend in PM10, then they claim that there is no reliable study on “tendencies” for particles lower than 1µm, after they support number measurements for Dp < 2.5 µm to establish health risk and “trends” (not clear of what type) of “pollution particles”.
The consistency and the clarity of this discussion is very limited, besides that the manuscript extensively describes vertical profile of aerosol mass concentration.
Moreover in the reply the authors state that “it is the first time that timeseries of vertical profile in urban conditions at ground and in altitude are available”. Unfortunately for the reader, the authors lack for fundamental bibliography about vertical profile of aerosols in urban atmosphere, nonetheless the hint provided during the first review. Now I would strongly recommend to add a brief overview of aerosol vertical profile in the Introduction of the revised manuscript to introduce the reader to the main findings and challenges in this type of data. Just to list a sample papers about aerosol vertical profiles: Ferrero et al. 2014 ACP with measurements in 3 sites in Italy, or Bisht et al Science of the Tot Environ 2016 in New Delhi or Liu et al Science of the Tot Environ 2020 in Macau, Rosati et al ACP 2016 (connected to a whole EU project “PEGASOS” on vertical profiling), and surely there are dozens more.
Line 156: decimals are indicated with a dot
Line 190: “Poison statistics”. Do the authors mean something related to the “Poisson distribution”?
To conclude, my recommendations are:
1. review the English language to avoid any misunderstanding (phrasing as “counting measurement” is still present)
2. include an overview of literature findings about aerosol vertical profiles either/or from balloons, UAVs, zeppelin, …
3. thoroughly discuss the limits/uncertainty of the results presented, due to the lack in the discussion of fundamental variables for the full understanding of the variability in the measured profiles (e.g. potential temperature). Note that all the main literature, including the examples cited above, include the variables describing atmospheric mixing.
Actually, even if these latter variables were not measured during the campaigns described in the present manuscript, they were surely simulated by the local weather forecast service and they are certainly available, or they can be recomputed; probably there are also atmospheric sounding data to serve the Paris airports. For future articles regarding these vertical profiles I would encourage you to include this data in the analysis.
Reviewer 3 Report
The author responded to my question. I think it is still feasible, so I modified my opinion. Especially the statement added by the author makes me feel that the conclusion is more credible. The whole article is more complete.
In addition, I have a little suggestion. The author should write a little more space in the meaning part to make the article more research-worthy.
